# Generalized Denoising Auto-Encoders as Generative Models

**Yoshua Bengio, Li Yao, Guillaume Alain, and Pascal Vincent**
Département d'informatique et recherche opérationnelle, Université de Montréal

## Abstract

Recent work has shown how denoising and contractive autoencoders implicitly capture the structure of the data-generating density, in the case where the corruption noise is Gaussian, the reconstruction error is the squared error, and the data is continuous-valued. This has led to various proposals for sampling from this implicitly learned density function, using Langevin and Metropolis-Hastings MCMC. However, it remained unclear how to connect the training procedure of regularized auto-encoders to the implicit estimation of the underlying data-generating distribution when the data are discrete, or using other forms of corruption process and reconstruction errors. Another issue is the mathematical justification which is only valid in the limit of small corruption noise. We propose here a different attack on the problem, which deals with all these issues: arbitrary (but noisy enough) corruption, arbitrary reconstruction loss (seen as a log-likelihood), handling both discrete and continuous-valued variables, and removing the bias due to non-infinitesimal corruption noise (or non-infinitesimal contractive penalty).

## 1   Introduction

Auto-encoders learn an encoder function from input to representation and a decoder function back from representation to input space, such that the reconstruction (composition of encoder and decoder) is good for training examples. Regularized auto-encoders also involve some form of regularization that prevents the auto-encoder from simply learning the identity function, so that reconstruction error will be low at training examples (and hopefully at test examples) but high in general. Different variants of auto-encoders and sparse coding have been, along with RBMs, among the most successful building blocks in recent research in deep learning (Bengio *et al.*, 2013b). Whereas the usefulness of auto-encoder variants as feature learners for supervised learning can directly be assessed by performing supervised learning experiments with unsupervised pre-training, what has remained until recently rather unclear is the interpretation of these algorithms in the context of pure unsupervised learning, as devices to capture the salient structure of the input data distribution. Whereas the answer is clear for RBMs, it is less obvious for regularized auto-encoders. Do they completely characterize the input distribution or only some aspect of it? For example, clustering algorithms such as k-means only capture the modes of the distribution, while manifold learning algorithms characterize the low-dimensional regions where the density concentrates.

Some of the first ideas about the probabilistic interpretation of auto-encoders were proposed by Ranzato *et al.* (2008): they were viewed as approximating an energy function through the reconstruction error, i.e., being trained to have low reconstruction error at the training examples and high reconstruction error elsewhere (through the regularizer, e.g., sparsity or otherwise, which prevents the auto-encoder from learning the identity function). An important breakthrough then came, yielding a first formal probabilistic interpretation of regularized auto-encoders as models of the input distribution, with the work of Vincent (2011). This work showed that some denoising auto-encoders (DAEs) correspond to a Gaussian RBM and that minimizing the denoising reconstruction error (as a squared error) estimates the energy function through a regularized form of score matching, with the regularization disappearing as the amount of corruption noise goes to 0, and then converging to the same solution as score matching (Hyvärinen, 2005). This connection and its generalization to other

energy functions, giving rise to the general denoising score matching training criterion, is discussed in several other papers (Kingma and LeCun, 2010; Swersky *et al.*, 2011; Alain and Bengio, 2013).

Another breakthrough has been the development of an empirically successful sampling algorithm for contractive auto-encoders (Rifai *et al.*, 2012), which basically involves composing encoding, decoding, and noise addition steps. This algorithm is motivated by the observation that the Jacobian matrix (of derivatives) of the encoding function provides an estimator of a local Gaussian approximation of the density, i.e., the leading singular vectors of that matrix span the tangent plane of the manifold near which the data density concentrates. However, a formal justification for this algorithm remains an open problem.

The last step in this development (Alain and Bengio, 2013) generalized the result from Vincent (2011) by showing that when a DAE (or a contractive auto-encoder with the contraction on the whole encode/decode reconstruction function) is trained with small Gaussian corruption and squared error loss, it estimates the score (derivative of the log-density) of the underlying data-generating distribution, which is proportional to the difference between reconstruction and input. This result does not depend on the parametrization of the auto-encoder, but suffers from the following limitations: it applies to one kind of corruption (Gaussian), only to continuous-valued inputs, only for one kind of loss (squared error), and it becomes valid only in the limit of small noise (even though in practice, best results are obtained with large noise levels, comparable to the range of the input).

What we propose here is a different probabilistic interpretation of DAEs, which is valid for any data type, any corruption process (so long as it has broad enough support), and any reconstruction loss (so long as we can view it as a log-likelihood).

The basic idea is that if we corrupt observed random variable $X$ into $\tilde{X}$ using conditional distribution $\mathcal{C}(\tilde{X}|X)$, we are really training the DAE to estimate the reverse conditional $P(X|\tilde{X})$. Combining this estimator with the known $\mathcal{C}(\tilde{X}|X)$, we show that we can recover a consistent estimator of $P(X)$ through a Markov chain that alternates between sampling from $P(X|\tilde{X})$ and sampling from $\mathcal{C}(\tilde{X}|X)$, i.e., encode/decode, sample from the reconstruction distribution model $P(X|\tilde{X})$, apply the stochastic corruption procedure $\mathcal{C}(\tilde{X}|X)$, and iterate.

This theoretical result is validated through experiments on artificial data in a non-parametric setting and experiments on real data in a parametric setting (with neural net DAEs). We find that we can improve the sampling behavior by using the model itself to define the corruption process, yielding a training procedure that has some surface similarity to the contrastive divergence algorithm (Hinton, 1999; Hinton *et al.*, 2006).

---

**Algorithm 1** THE GENERALIZED DENOISING AUTO-ENCODER TRAINING ALGORITHM *requires a training set or training distribution $\mathcal{D}$ of examples $X$, a given corruption process $\mathcal{C}(\tilde{X}|X)$ from which one can sample, and with which one trains a conditional distribution $P_\theta(X|\tilde{X})$ from which one can sample.*

---

**repeat**
- sample training example $X \sim \mathcal{D}$
- sample corrupted input $\tilde{X} \sim \mathcal{C}(\tilde{X}|X)$
- use $(X, \tilde{X})$ as an additional training example towards minimizing the expected value of $-\log P_\theta(X|\tilde{X})$, e.g., by a gradient step with respect to $\theta$.

**until** convergence of training (e.g., as measured by early stopping on out-of-sample negative log-likelihood)

---

## 2 Generalizing Denoising Auto-Encoders

### 2.1 Definition and Training

Let $\mathcal{P}(X)$ be the data-generating distribution over observed random variable $X$. Let $\mathcal{C}$ be a given corruption process that stochastically maps an $X$ to a $\tilde{X}$ through conditional distribution $\mathcal{C}(\tilde{X}|X)$. The training data for the generalized denoising auto-encoder is a set of pairs $(X, \tilde{X})$ with $X \sim \mathcal{P}(X)$ and $\tilde{X} \sim \mathcal{C}(\tilde{X}|X)$. The DAE is trained to predict $X$ given $\tilde{X}$ through a learned conditional distribution $P_\theta(X|\tilde{X})$, by choosing this conditional distribution within some family of distributions

indexed by $\theta$, not necessarily a neural net. The training procedure for the DAE can generally be formulated as learning to predict $X$ given $\tilde{X}$ by possibly regularized maximum likelihood, i.e., the generalization performance that this training criterion attempts to minimize is

$$\mathcal{L}(\theta) = -E[\log P_\theta(X|\tilde{X})] \qquad (1)$$

where the expectation is taken over the joint data-generating distribution
$$\mathcal{P}(X, \tilde{X}) = \mathcal{P}(X)\mathcal{C}(\tilde{X}|X). \qquad (2)$$

## 2.2 Sampling

We define the following pseudo-Gibbs Markov chain associated with $P_\theta$:

$$X_t \sim P_\theta(X|\tilde{X}_{t-1})$$
$$\tilde{X}_t \sim \mathcal{C}(\tilde{X}|X_t) \qquad (3)$$

which can be initialized from an arbitrary choice $X_0$. This is the process by which we are going to generate samples $X_t$ according to the model implicitly learned by choosing $\theta$. We define $T(X_t|X_{t-1})$ the transition operator that defines a conditional distribution for $X_t$ given $X_{t-1}$, independently of $t$, so that the sequence of $X_t$'s forms a homogeneous Markov chain. If the asymptotic marginal distribution of the $X_t$'s exists, we call this distribution $\pi(X)$, and we show below that it consistently estimates $\mathcal{P}(X)$.

Note that the above chain is not a proper Gibbs chain in general because there is no guarantee that $P_\theta(X|\tilde{X}_{t-1})$ and $\mathcal{C}(\tilde{X}|X_t)$ are consistent with a unique joint distribution. In that respect, the situation is similar to the sampling procedure for dependency networks (Heckerman *et al.*, 2000), in that the pairs $(X_t, \tilde{X}_{t-1})$ are not guaranteed to have the same asymptotic distribution as the pairs $(X_t, \tilde{X}_t)$ as $t \to \infty$. As a follow-up to the results in the next section, it is shown in Bengio *et al.* (2013a) that dependency networks can be cast into the same framework (which is that of Generative Stochastic Networks), and that if the Markov chain is ergodic, then its stationary distribution will define a joint distribution between the random variables (here that would be $X$ and $\tilde{X}$), even if the conditionals are not consistent with it.

## 2.3 Consistency

Normally we only have access to a finite number $n$ of training examples but as $n \to \infty$, the empirical training distribution approaches the data-generating distribution. To compensate for the finite training set, we generally introduce a (possibly data-dependent) regularizer $\Omega$ and the actual training criterion is a sum over $n$ training examples $(X, \tilde{X})$,

$$\mathcal{L}_n(\theta) = \frac{1}{n} \sum_{X \sim \mathcal{P}(X), \tilde{X} \sim \mathcal{C}(\tilde{X}|X)} \lambda_n \Omega(\theta, X, \tilde{X}) - \log P_\theta(X|\tilde{X}) \qquad (4)$$

where we allow the regularization coefficient $\lambda_n$ to be chosen according to the number of training examples $n$, with $\lambda_n \to 0$ as $n \to \infty$. With $\lambda_n \to 0$ we get that $\mathcal{L}_n \to \mathcal{L}$ (i.e. converges to generalization error, Eq. 1), so consistent estimators of $\mathcal{P}(X|\tilde{X})$ stay consistent. We define $\theta_n$ to be the minimizer of $\mathcal{L}_n(\theta)$ when given $n$ training examples.

We define $T_n$ to be the transition operator $T_n(X_t|X_{t-1}) = \int P_{\theta_n}(X_t|\tilde{X})\mathcal{C}(\tilde{X}|X_{t-1})d\tilde{X}$ associated with $\theta_n$ (the parameter obtained by minimizing the training criterion with $n$ examples), and define $\pi_n$ to be the asymptotic distribution of the Markov chain generated by $T_n$ (if it exists). We also define $T$ be the operator of the Markov chain associated with the learned model as $n \to \infty$.

**Theorem 1.** *If $P_{\theta_n}(X|\tilde{X})$ is a consistent estimator of the true conditional distribution $\mathcal{P}(X|\tilde{X})$ and $T_n$ defines an ergodic Markov chain, then as the number of examples $n \to \infty$, the asymptotic distribution $\pi_n(X)$ of the generated samples converges to the data-generating distribution $\mathcal{P}(X)$.*

*Proof.* If $T_n$ is ergodic, then the Markov chain converges to a $\pi_n$. Based on our definition of the "true" joint (Eq. 2), one obtains a conditional $\mathcal{P}(X|\tilde{X}) \propto \mathcal{P}(X)\mathcal{C}(\tilde{X}|X)$. This conditional, along with $\mathcal{P}(\tilde{X}|X) = \mathcal{C}(\tilde{X}|X)$ can be used to define a proper Gibbs chain where one alternatively samples from $\mathcal{P}(\tilde{X}|X)$ and from $\mathcal{P}(X|\tilde{X})$. Let $\mathcal{T}$ be the corresponding "true" transition operator, which maps the $t$-th sample $X$ to the $t+1$-th in that chain. That is, $\mathcal{T}(X_t|X_{t-1}) = \int \mathcal{P}(X_t|\tilde{X})\mathcal{C}(\tilde{X}|X_{t-1})d\tilde{X}$. $\mathcal{T}$ produces $\mathcal{P}(X)$ as asymptotic marginal distribution over $X$ (as we

consider more samples from the chain) simply because $\mathcal{P}(X)$ is the marginal distribution of the joint $\mathcal{P}(X)\mathcal{C}(\tilde{X}|X)$ to which the chain converges. By hypothesis we have that $P_{\theta_n}(X|\tilde{X}) \to \mathcal{P}(X|\tilde{X})$ as $n \to \infty$. Note that $T_n$ is defined exactly as $\mathcal{T}$ but with $\mathcal{P}(X_t|\tilde{X})$ replaced by $P_{\theta_n}(X|\tilde{X})$. Hence $T_n \to \mathcal{T}$ as $n \to \infty$.

Now let us convert the convergence of $T_n$ to $\mathcal{T}$ into the convergence of $\pi_n(X)$ to $\mathcal{P}(X)$. We will exploit the fact that for the 2-norm, matrix $M$ and unit vector $v$, $||Mv||_2 \leq \sup_{||x||_2=1} ||Mx||_2 = ||M||_2$. Consider $M = \mathcal{T} - T_n$ and $v$ the principal eigenvector of $\mathcal{T}$, which, by the Perron-Frobenius theorem, corresponds to the asymptotic distribution $\mathcal{P}(X)$. Since $T_n \to \mathcal{T}$, $||\mathcal{T} - T_n||_2 \to 0$. Hence $||(\mathcal{T} - T_n)v||_2 \leq ||\mathcal{T} - T_n||_2 \to 0$, which implies that $T_n v \to \mathcal{T} v = v$, where the last equality comes from the Perron-Frobenius theorem (the leading eigenvalue is 1). Since $T_n v \to v$, it implies that $v$ becomes the leading eigenvector of $T_n$, i.e., the asymptotic distribution of the Markov chain, $\pi_n(X)$ converges to the true data-generating distribution, $\mathcal{P}(X)$, as $n \to \infty$. $\qquad\square$

Hence the asymptotic sampling distribution associated with the Markov chain defined by $T_n$ (i.e., the model) implicitly defines the distribution $\pi_n(X)$ learned by the DAE over the observed variable $X$. Furthermore, that estimator of $\mathcal{P}(X)$ is consistent so long as our (regularized) maximum likelihood estimator of the conditional $P_\theta(X|\tilde{X})$ is also consistent. We now provide sufficient conditions for the ergodicity of the chain operator (i.e. to apply theorem 1).

**Corollary 1.** **If** $P_\theta(X|\tilde{X})$ *is a consistent estimator of the true conditional distribution $\mathcal{P}(X|\tilde{X})$,* **and** *both the data-generating distribution and denoising model are contained in and non-zero in a finite-volume region $V$ (i.e., $\forall \tilde{X}$, $\forall X \notin V$, $\mathcal{P}(X) = 0, P_\theta(X|\tilde{X}) = 0$),* **and** $\forall \tilde{X}$, $\forall X \in V$, $\mathcal{P}(X) > 0$, $P_\theta(X|\tilde{X}) > 0, \mathcal{C}(\tilde{X}|X) > 0$ **and** *these statements remain true in the limit of $n \to \infty$,* **then** *the asymptotic distribution $\pi_n(X)$ of the generated samples converges to the data-generating distribution $\mathcal{P}(X)$.*

*Proof.* To obtain the existence of a stationary distribution, it is sufficient to have irreducibility (every value reachable from every other value), aperiodicity (no cycle where only paths through the cycle allow to return to some value), and recurrence (probability 1 of returning eventually). These conditions can be generalized to the continuous case, where we obtain ergodic Harris chains rather than ergodic Markov chains. If $P_\theta(X|\tilde{X}) > 0$ and $\mathcal{C}(\tilde{X}|X) > 0$ (for $X \in V$), then $T_n(X_t|X_{t-1}) > 0$ as well, because

$$T(X_t|X_{t-1}) = \int P_\theta(X_t|\tilde{X})\mathcal{C}(\tilde{X}|X_{t-1})d\tilde{X}$$

This positivity of the transition operator guarantees that one can jump from any point in $V$ to any other point in one step, thus yielding *irreducibility* and *aperiodicity*. To obtain *recurrence* (preventing the chain from diverging to infinity), we rely on the assumption that the domain $V$ is bounded. Note that although $T_n(X_t|X_{t-1}) > 0$ could be true for any finite $n$, we need this condition to hold for $n \to \infty$ as well, to obtain the consistency result of theorem 1. By assuming this positivity (Boltzmann distribution) holds for the data-generating distribution, we make sure that $\pi_n$ does not converge to a distribution which puts 0's anywhere in $V$. Having satisfied all the conditions for the existence of a stationary distribution for $T_n$ as $n \to \infty$, we can apply theorem 1 and obtain its conclusion. $\qquad\square$

Note how these conditions take care of the various troubling cases one could think of. We avoid the case where there is no corruption (which would yield a wrong estimation, with the DAE simply learning a dirac probability its input). Second, we avoid the case where the chain wanders to infinity by assuming a finite volume where the model and data live, a real concern in the continuous case. If it became a real issue, we could perform rejection sampling to make sure that $P(X|\tilde{X})$ produces $X \in V$.

## 2.4 Locality of the Corruption and Energy Function

If we believe that $P(X|\tilde{X})$ is well estimated for all $(X, \tilde{X})$ pairs, i.e., that it is approximately consistent with $\mathcal{C}(\tilde{X}|X)$, then we get as many estimators of the energy function as we want, by picking a particular value of $\tilde{X}$.

Let us define the notation $P(\cdot)$ to denote the probability of the joint, marginals or conditionals over the pairs $(X_t, \tilde{X}_{t-1})$ that are produced by the model's Markov chain $T$ as $t \to \infty$. So $P(X) = \pi(X)$

is the asymptotic distribution of the Markov chain $T$, and $P(\tilde{X})$ the marginal over the $\tilde{X}$'s in that chain. The above assumption means that $P(\tilde{X}_{t-1}|X_t) \approx \mathcal{C}(\tilde{X}_{t-1}|X_t)$ (which is not guaranteed in general, but only asymptotically as $P$ approaches the true $\mathcal{P}$). Then, by Bayes rule, $P(X) = \frac{P(X|\tilde{X})P(\tilde{X})}{P(\tilde{X}|X)} \approx \frac{P(X|\tilde{X})P(\tilde{X})}{\mathcal{C}(\tilde{X}|X)} \propto \frac{P(X|\tilde{X})}{\mathcal{C}(\tilde{X}|X)}$ so that we can get an estimated energy function from any given choice of $\tilde{X}$ through $\mathrm{energy}(X) \approx -\log P(X|\tilde{X}) + \log \mathcal{C}(\tilde{X}|X)$. where one should note that the intractable *partition function depends on the chosen value of $\tilde{X}$*.

How much can we trust that estimator and how should $\tilde{X}$ be chosen? First note that $P(X|\tilde{X})$ has only been trained for pairs $(X, \tilde{X})$ for which $\tilde{X}$ is relatively close to $X$ (assuming that the corruption is indeed changing $X$ generally into some neighborhood). Hence, although in theory (with infinite amount of data and capacity) the above estimator should be good, in practice it might be poor when $X$ is far from $\tilde{X}$. So if we pick a particular $\tilde{X}$ the estimated energy might be good for $X$ in the neighborhood of $\tilde{X}$ but poor elsewhere. What we could do though, is use a different approximate energy function in different regions of the input space. Hence the above estimator gives us a way to compare the probabilities of nearby points $X_1$ and $X_2$ (through their difference in energy), picking for example a midpoint $\tilde{X} = \frac{1}{2}(X_1 + X_2)$. One could also imagine that if $X_1$ and $X_N$ are far apart, we could chart a path between $X_1$ and $X_N$ with intermediate points $X_k$ and use an estimator of the relative energies between the neighbors $X_k, X_{k+1}$, add them up, and obtain an estimator of the relative energy between $X_1$ and $X_N$.

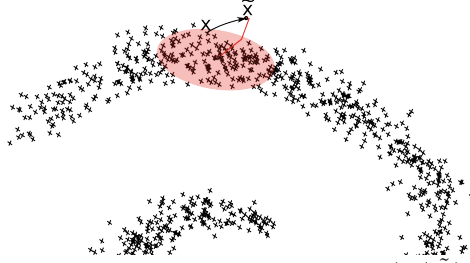

Figure 1: *Although $\mathcal{P}(X)$ may be complex and multi-modal, $\mathcal{P}(X|\tilde{X})$ is often simple and approximately unimodal (e.g., multivariate Gaussian, pink oval) for most values of $\tilde{X}$ when $\mathcal{C}(\tilde{X}|X)$ is a local corruption. $\mathcal{P}(X)$ can be seen as an infinite mixture of these local distributions (weighted by $\mathcal{P}(\tilde{X})$).*

This brings up an interesting point. If we could always obtain a good estimator $P(X|\tilde{X})$ for any $\tilde{X}$, we could just train the model with $\mathcal{C}(\tilde{X}|X) = \mathcal{C}(\tilde{X})$, i.e., with an unconditional noise process that ignores $X$. In that case, the estimator $P(X|\tilde{X})$ would directly equal $P(X)$ since $\tilde{X}$ and $X$ are actually sampled independently in its "denoising" training data. We would have gained nothing over just training any probabilistic model just directly modeling the observed $X$'s. The gain we expect from using the denoising framework is that if $\tilde{X}$ is a local perturbation of $X$, then the true $\mathcal{P}(X|\tilde{X})$ can be well approximated by a much simpler distribution than $\mathcal{P}(X)$. See Figure 1 for a visual explanation: in the limit of very small perturbations, one could even assume that $\mathcal{P}(X|\tilde{X})$ can be well approximated by a simple unimodal distribution such as the Gaussian (for continuous data) or factorized binomial (for discrete binary data) commonly used in DAEs as the reconstruction probability function (conditioned on $\tilde{X}$). This idea is already behind the non-local manifold Parzen windows (Bengio *et al.*, 2006a) and non-local manifold tangent learning (Bengio *et al.*, 2006b) algorithms: the local density around a point $\tilde{X}$ can be approximated by a multivariate Gaussian whose covariance matrix has leading eigenvectors that *span the local tangent of the manifold* near which the data concentrates (if it does). The idea of a locally Gaussian approximation of a density with a manifold structure is also exploited in the more recent work on the contractive auto-encoder (Rifai *et al.*, 2011) and associated sampling procedures (Rifai *et al.*, 2012). Finally, strong theoretical evidence in favor of that idea comes from the result from Alain and Bengio (2013): *when the amount of corruption noise converges to 0 and the input variables have a smooth continuous density, then a unimodal Gaussian reconstruction density suffices to fully capture the joint distribution.*

Hence, although $P(X|\tilde{X})$ encapsulates all information about $\mathcal{P}(X)$ (assuming $\mathcal{C}$ given), it will generally have far fewer non-negligible modes, making easier to approximate it. This can be seen analytically by considering the case where $\mathcal{P}(X)$ is a mixture of many Gaussians and the corruption

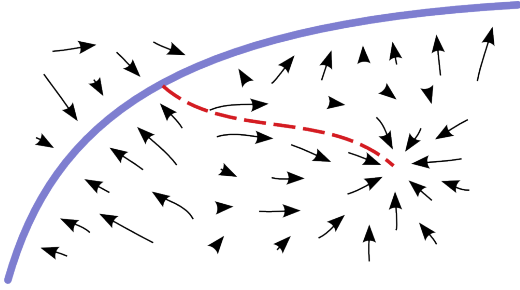

Figure 2: Walkback samples get attracted by spurious modes and contribute to removing them. Segment of data manifold in violet and example walkback path in red dotted line, starting on the manifold and going towards a spurious attractor. The vector field represents expected moves of the chain, for a unimodal $P(X|\tilde{X})$, with arrows from $\tilde{X}$ to $X$.

is a local Gaussian: $P(X|\tilde{X})$ remains a Gaussian mixture, but one for which most of the modes have become negligible (Alain and Bengio, 2013). We return to this in Section 3, suggesting that in order to avoid spurious modes, it is better to have non-infinitesimal corruption, allowing faster mixing and successful burn-in not pulled by spurious modes far from the data.

## 3 Reducing the Spurious Modes with Walkback Training

Sampling in high-dimensional spaces (like in experiments below) using a simple local corruption process (such as Gaussian or salt-and-pepper noise) suggests that if the corruption is too local, the DAE's behavior far from the training examples can create spurious modes in the regions insufficiently visited during training. More training iterations or increasing the amount of corruption noise helps to substantially alleviate that problem, but we discovered an even bigger boost by *training the DAE Markov chain to walk back towards the training examples* (see Figure 2). We exploit knowledge of the currently learned model $P(X|\tilde{X})$ to define the corruption, so as to pick values of $\tilde{X}$ that would be obtained by following the generative chain: wherever the model would go if we sampled using the generative Markov chain starting at a training example $X$, we consider to be a kind of "negative example" $\tilde{X}$ from which the auto-encoder should move away (and towards $X$). The spirit of this procedure is thus very similar to the CD-$k$ (Contrastive Divergence with $k$ MCMC steps) procedure proposed to train RBMs (Hinton, 1999; Hinton *et al.*, 2006).

More precisely, the modified corruption process $\tilde{\mathcal{C}}$ we propose is the following, based on the original corruption process $\mathcal{C}$. We use it in a version of the training algorithm called **walkback**, where we replace the corruption process $\mathcal{C}$ of Algorithm 1 by the walkback process $\tilde{\mathcal{C}}$ of Algorithm 2. This also provides extra training examples (taking advantage of the $\tilde{X}$ samples generated along the walk away from $X$). It is called **walkback** because it forces the DAE to learn to walk back from the random walk it generates, towards the $X$'s in the training set.

Algorithm 2: THE WALKBACK ALGORITHM *is based on the walkback corruption process $\tilde{\mathcal{C}}(\tilde{X}|X)$, defined below in terms of a generic original corruption process $\mathcal{C}(\tilde{X}|X)$ and the current model's reconstruction conditional distribution $P(X|\tilde{X})$. For each training example $X$, it provides a sequence of additional training examples $(X, \tilde{X}^*)$ for the DAE. It has a hyper-parameter that is a geometric distribution parameter $0 < p < 1$ controlling the length of these walks away from $X$, with $p = 0.5$ by default. Training by Algorithm 1 is the same, but using all $\tilde{X}^*$ in the returned list $L$ to form the pairs $(X, \tilde{X}^*)$ as training examples instead of just $(X, \tilde{X})$.*

1:  $X^* \leftarrow X, L \leftarrow [\,]$
2:  Sample $\tilde{X}^* \sim \mathcal{C}(\tilde{X}|X^*)$
3:  Sample $u \sim \text{Uniform}(0, 1)$
4:  **if** $u > p$ **then**
5:     Append $\tilde{X}^*$ to $L$ and **return** $L$
6:  If during training, append $\tilde{X}^*$ to $L$, so $(X, \tilde{X}^*)$ will be an additional training example.
7:  Sample $X^* \sim P(X|\tilde{X}^*)$
8:  **goto** 2.

**Proposition 1.** *Let $P(X)$ be the implicitly defined asymptotic distribution of the Markov chain alternating sampling from $P(X|\tilde{X})$ and $\mathcal{C}(\tilde{X}|X)$, where $\mathcal{C}$ is the original local corruption process. Under the assumptions of corollary 1, minimizing the training criterion in walkback training algo-*

*rithm for generalized DAEs (combining Algorithms 1 and 2) produces a $P(X)$ that is a consistent estimator of the data generating distribution $\mathcal{P}(X)$.*

*Proof.* Consider that during training, we produce a sequence of estimators $P_k(X|\tilde{X})$ where $P_k$ corresponds to the $k$-th training iteration (modifying the parameters after each iteration). With the walkback algorithm, $P_{k-1}$ is used to obtain the corrupted samples $\tilde{X}$ from which the next model $P_k$ is produced. If training converges, $P_k \approx P_{k+1} = P$ and we can then consider the whole corruption process $\tilde{\mathcal{C}}$ fixed. By corollary 1, the Markov chain obtained by alternating samples from $P(X|\tilde{X})$ and samples from $\tilde{\mathcal{C}}(\tilde{X}|X)$ converges to an asymptotic distribution $P(X)$ which estimates the underlying data-generating distribution $\mathcal{P}(X)$. The walkback corruption $\tilde{\mathcal{C}}(\tilde{X}|X)$ corresponds to a few steps alternating sampling from $\mathcal{C}(\tilde{X}|X)$ (the fixed local corruption) and sampling from $P(X|\tilde{X})$. Hence the overall sequence when using $\tilde{\mathcal{C}}$ can be seen as a Markov chain obtained by alternatively sampling from $\mathcal{C}(\tilde{X}|X)$ and from $P(X|\tilde{X})$ just as it was when using merely $\mathcal{C}$. Hence, once the model is trained with walkback, one can sample from it usingc orruption $\mathcal{C}(\tilde{X}|X)$. $\qquad\square$

A consequence is that *the walkback training algorithm estimates the same distribution as the original denoising algorithm*, but may do it more efficiently (as we observe in the experiments), by exploring the space of corruptions in a way that spends more time where it most helps the model.

## 4  Experimental Validation

**Non-parametric case.** The mathematical results presented here apply to any denoising training criterion where the reconstruction loss can be interpreted as a negative log-likelihood. This remains true whether or not the denoising machine $P(X|\tilde{X})$ is parametrized as the composition of an encoder and decoder. This is also true of the asymptotic estimation results in Alain and Bengio (2013). We experimentally validate the above theorems in a case where the asymptotic limit (of enough data and enough capacity) can be reached, i.e., in a low-dimensional non-parametric setting. Fig. 3 shows the distribution recovered by the Markov chain for **discrete data** with only 10 different values. The conditional $P(X|\tilde{X})$ was estimated by multinomial models and maximum likelihood (counting) from 5000 training examples. 5000 samples were generated from the chain to estimate the asymptotic distribution $\pi_n(X)$. For **continuous data**, Figure 3 also shows the result of 5000 generated samples and 500 original training examples with $X \in \mathbf{R}^{10}$, with scatter plots of pairs of dimensions. The estimator is also non-parametric

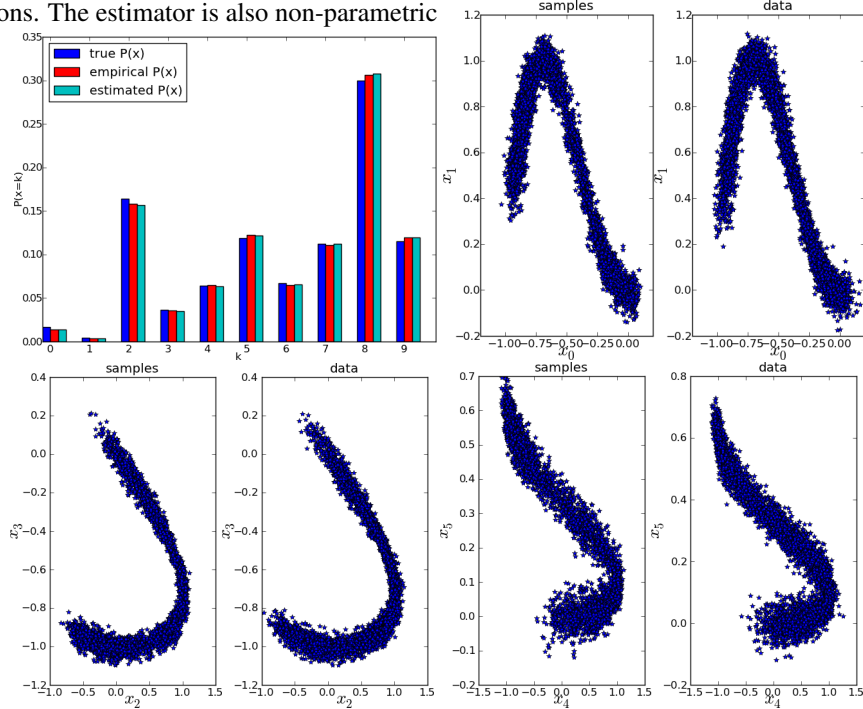

Figure 3: *Top left: histogram of a data-generating distribution (true, blue), the empirical distribution (red), and the estimated distribution using a denoising maximum likelihood estimator. Other figures: pairs of variables (out of 10) showing the training samples and the model-generated samples.*

**MNIST digits.** We trained a DAE on the binarized MNIST data (thresholding at 0.5). A Theano[1] (Bergstra *et al.*, 2010) implementation is available[2]. The 784-2000-784 auto-encoder is trained for 200 epochs with the 50000 training examples and salt-and-pepper noise (probability 0.5 of corrupting each bit, setting it to 1 or 0 with probability 0.5). It has 2000 tanh hidden units and is trained by minimizing cross-entropy loss, i.e., maximum likelihood on a factorized Bernoulli reconstruction distribution. With walkback training, a chain of 5 steps was used to generate 5 corrupted examples for each training example. Figure 4 shows samples generated with and without walkback. The quality of the samples was also estimated quantitatively by measuring the log-likelihood of the test set under a non-parametric density estimator $\hat{P}(x) = \text{mean}_{\tilde{X}} P(x|\tilde{X})$ constructed from 10000 consecutively generated samples ($\tilde{X}$ from the Markov chain). The expected value of $E[\hat{P}(x)]$ over the samples can be shown (Bengio and Yao, 2013) to be a lower bound (i.e. conservative estimate) of the true (implicit) model density $P(x)$. The test set log-likelihood bound was not used to select among model architectures, but visual inspection of samples generated did guide the preliminary search reported here. Optimization hyper-parameters (learning rate, momentum, and learning rate reduction schedule) were selected based on the training objective. We compare against a state-of-the-art RBM (Cho *et al.*, 2013) with an AIS log-likelihood estimate of -64.1 (AIS estimates tend to be optimistic). We also drew samples from the RBM and applied the same estimator (using the mean of the RBM's $P(x|h)$ with $h$ sampled from the Gibbs chain), and obtained a log-likelihood non-parametric bound of -233, skipping 100 MCMC steps between samples (otherwise numbers are very poor for the RBM, which does not mix at all). The DAE log-likelihood bound with and without walkback is respectively -116 and -142, confirming visual inspection suggesting that the walkback algorithm produces less spurious samples. However, the RBM samples can be improved by a spatial blur. By tuning the amount of blur (the spread of the Gaussian convolution), we obtained a bound of -112 for the RBM. Blurring did not help the auto-encoder.

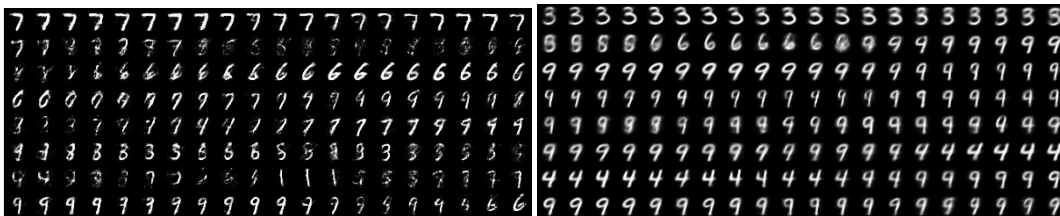

Figure 4: *Successive samples generated by Markov chain associated with the trained DAEs according to the plain sampling scheme (left) and walkback sampling scheme (right). There are less "spurious" samples with the walkback algorithm.*

## 5    Conclusion and Future Work

We have proven that training a model to denoise is a way to implicitly estimate the underlying data-generating process, and that a simple Markov chain that alternates sampling from the denoising model and from the corruption process converges to that estimator. This provides a means for generating data from any DAE (if the corruption is not degenerate, more precisely, if the above chain converges). We have validated those results empirically, both in a non-parametric setting and with real data. This study has also suggested a variant of the training procedure, *walkback training*, which seem to converge faster to same the target distribution.

One of the insights arising out of the theoretical results presented here is that in order to reach the asymptotic limit of fully capturing the data distribution $\mathcal{P}(X)$, it may be necessary for the model's $P(X|\tilde{X})$ to have the ability to represent multi-modal distributions over $X$ (given $\tilde{X}$).

### Acknowledgments

The authors would acknowledge input from A. Courville, I. Goodfellow, R. Memisevic, K. Cho as well as funding from NSERC, CIFAR (YB is a CIFAR Fellow), and Canada Research Chairs.

## Footnotes

[1]http://deeplearning.net/software/theano/

[2]git@github.com:yaoli/GSN.git

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
