[Reviews · NeurIPS 2013]

Submitted by Assigned_Reviewer_5

This paper continues a recent line of theoretical work that seeks to explain what autoencoders learn about the data-generating distribution. Of practical importance from this work have been ways to sample from autoencoders. Specifically, this paper picks up where (Alain and Bengio 2013) left off. That paper was able to show that autoencoders (under a number of conditions) estimate the score (derivative of the log-density) of the data-generating distribution in a way that was proportional to the difference between reconstruction and input. However, it was these conditions that limited this work: it only considered Gaussian corruption, it only applied to continuous inputs, it was proven for only squared error, and was valid only in the limit of small corruption. The current paper connects the autoencoder training procedure to the implicit estimation of the data-generating distribution for arbitrary corruption, arbitrary reconstruction loss, and can handle both discrete and continuous variables for non-infinitesimal corruption noise. Moreover, the paper presents a new training algorithm called "walkback" which estimates the same distribution as the "vanilla" denoising algorithm, but, as experimental evidence suggests, may do so in a more efficient way.

Quality: I'm not much of a theorist, but I enjoyed reading this paper as I did the predecessor (Alain and Bengio 2013). As I stated earlier, the theoretical work on understanding autoencoders has produced useful sampling schemes, and the walkback training algorithm, though not rigorously shown to be effective, may also have practical implications.

Clarity: The paper does a nice job of reviewing the literature around the probabilistic interpretation of autoencoders. It situates itself well in the context of this other work, and makes clear its contributions. The generalized denoising algorithm (Algorithm 1) is intuitively presented before the more theoretical discussion in Section 2. In short, it's a very simple, elegant generalization. I followed the thrust of the theory (Theorem 1, Corollary 1) but I haven't checked for complete correctness. Section 2.4 provides a nice insight that motivates the denoising framework: if X^tilde is a "local" corruption of X, then the true "reconstruction" distribution P(X|X^tilde) can be well approximated by a much simpler distribution than the true P(X).

Originality: The work concentrates on a fairly specific model class, and the theory is novel.

Significance: See my comments above re: practical significance (sampling) and potential implications of walkback algorithm. Widespread impact may depend on more thorough experimental validation.

Comments

- In the section that introduces "Walkback", present Algorithm 2 before the theory (Proposition 1), so that the reader has a clear idea of what walkback entails (even if it is simple!)

- I thought that the description of the non-parametric case of the experimental validation could be improved (in terms of clarity)

- Algorithm 2 gives a stochastic stopping criterion for walkback, but the experiments seem to imply that it is always stopped after 5 steps. Can you clarify this?
Summary: Interesting theoretical work on denoising autoencoders that removes many limitations of recent theory on this subject. The proposed "walkback" learning algorithm may also have practical importance.

Submitted by Assigned_Reviewer_6

This paper develops a step towards the first useful formulation of a denoising autoencoder, a particular flavor of neural network which implicitly captures the structure of the data by reproducing the input at its output, as a generative model - i.e. as a probabilistic model defining a distribution over the data. In particular, this paper develops a methodology for drawing samples from the data distribution implicitly captured by the denoising autoencoder. The paper does not develop a formulation of denoising autoencoders as generative models directly nor a method to extract the probability under the model of a novel data case. Nevertheless, this is a step towards the solution to a longstanding problem in the autoencoder and deep learning literature. Restricted Boltzmann machines, have existed as generative models of the data for many years but have remained difficult to use as such in practice, particularly for non-binary valued data. Thus, this work may offer a useful alternative for continuous valued data. From the revelations of the paper, the authors develop a new training algorithm for denoising autoencoders which they claim is better than existing methods - this is however not empirically validated. The paper is well written, structured and clear. The methodology and formulation appears to be correct and the paper offers some interesting discussion and intuition. The empirical validation is somewhat weak given the claims made in the paper. For example, the empirical justification of the walkback algorithm introduced in the paper is purely qualitative and only on one example. Given the wide practical applicability of denoising autoencoders, and a vast literature of corresponding quantitative empirical benchmarks, it seems strange to not see a quantitative comparison between denoising autoencoders trained with the walkback algorithm compared to those without. Also, the only quantitative empirical result (line 404) would suggest that the walkback algorithm in fact does not improve (quantitatively) at all over the standard procedure despite some additional cost. This is unfortunate as it seems that the walkback algorithm should really be empirically validated for the claim that it reduces spurious modes to be justified. Thus it seems unclear whether the sampling algorithm introduced in the paper can be used to actually improve autoencoder training.

On line 350, the authors state that they observe in the empirical evaluation that the walkback algorithm is more efficient than the standard training algorithm - however, this is nowhere to be found in the empirical evaluation section.

Quality:
The overall quality of ideas, writing and concepts is high. The empirical evaluation could be significantly more thorough.

Clarity:
The paper is extremely clear, well written and structured.

Originality:
The work is certainly original and attempts to address a heretofore unsolved problem with denoising autoencoders. The algorithm for sampling from the data density captured by the autoencoder is neat and novel.

Significance:
This is a possibly significant step towards solving a longstanding problem in the autoencoder literature. However, some of the theoretical aspects of the paper are not sufficiently empirically validated. It is especially not clear from the empirical validation whether the 'walkback algorithm' is actually practically useful. The ability to draw samples from the data distribution is very neat but it would be useful to have some significant empirical justification of the practical utility of this. Much remains to be solved to use denoising autoencoders as generative models - thus the title may be somewhat misleading. It seems the paper claims much more than is actually justified by empirical evaluation.
Summary: This paper makes a step towards formulating denoising autoencoders as generative models in that a methodology is developed to draw samples from the data distribution. This is a neat and novel contribution to the autoencoder literature but much remains to be done to actually formulate an autoencoder as a proper generative model (and be able to, for example, evaluate the probability of a novel data example under the model). The paper develops a novel training scheme called the 'walkback algorithm' but the algorithm is not empirically justified in any compelling manner. The paper is very clear and very well written but lacks substantial empirical evaluation to justify its claims.

Submitted by Assigned_Reviewer_7

Summary

This paper proposes a new probabilistic interpretation and two variants of a new algorithm for training denoising auto-encoders which removes restrictions of data type (sparse), corruption process, and reconstruction loss that were assumptions of earlier work. The originality of the proposed algorithms is good. The paper is very well written and interesting. The paper's clarity is also very good. The paper presents a very nice theoretical argument for the new algorithm and its walkback variant. A little more high level background would help improve the paper for non-experts. While, the experimental validation results are encouraging, they seem preliminary and not conclusive with respect to improvements over existing work. Given this limitation, it is difficult to predict the long term significance of the work.

Review Details
Line 20, in the abstract talks about sampling usin g MCMC … for what purpose?

Line 79. Why is there a distinction between the C (i.e. C(X^tilde|X) and the P (i.e. P(X|X^tilde)? Aren’t they both conditional distributions? Does C stand for corruption? If so, adding a clarification sentence at this point would help. Overall, I think this key paragraph could use a bit more text to frame the problem.

Line 81, How valid is the assumption that C(X^tilde|X) is known? This seems like a very restrictive assumption. Are the authors suggesting that they specify C(X^tilde|X) up front? If so, they should provide some examples of different types of corruptions that are applicable in this case. A simple high-level figure and explanation would be really helpful for most readers.

Line 83, isn’t sampling from P(X|X_tilde) then from C(X_tilde|X) sampling from decode and encode, not encode and decode as written in the paper?

“Algorithm 1” is specified in section 1, but not referenced in the paper until section 3.

How long did it take to train the new denoising auto-encoders with the plain and walkback sampling schemes?

The authors compare the algorithms with plain and walkback sampling to the algorithm in Bengio (2013a) (which they beat) on MNIST. They do not compare the results to any other previously proposed algorithms. The experimental validation section is definitely the weakest aspect of the paper. The results are encouraging, but not conclusive with respect to prior work. Without further studies, it is difficult to assess the potential impact of the work.

Typos
Pg. 3, line 156 “define T be” -> “define T to be”
Pg. 4, line 165, “T produces P(X) as asymptotic”
Pg. 4, line 214, “a dirac probability its input”
Line 229, extra period.
Line 314 “training criterion in walkback training algorithm”
Summary: This paper, which proposes a new probabilistic interpretation and two variants of a new algorithm for training denoising auto-encoders which removes restrictions of data type (sparse), corruption process, and reconstruction loss that were assumptions of earlier work, is well written and interesting. While, the experimental validation results are encouraging, they seem preliminary and not conclusive with respect to improvements over existing work.
Author Feedback

Author rebuttal: Rev5

> "The work concentrates on a fairly specific model class" ... "Significance"

The theory is completely agnostic about model class. It is more about a training criterion, showing that it allows to capture the data generating distribution, for any consistent parametrizations of a Markov chain transition operator. It is a radical departure from traditional probabilistic modeling (parametrizing P(x) explicitly): instead one parametrizes one step of a Markov chain that will generate the desired distribution. The theorem is validated empirically and specifies how to train, and the beauty of it is that the gradient can be computed easily by simple backprop, without requiring any variational or MCMC approximation *to get the gradient*, like in most graphical model formulations. The fundamental reason parametrizing a distribution through its generative chain is interesting is because the conditional distribution will tend to have less major modes, because moves are typically local, making the partition function gradient of this conditional distribution much easier to estimate.

> "Algorithm 2 ... experiments seem to imply that it is always stopped after 5 steps. Can you clarify?"

Any stopping criterion for the number of steps is compatible with the theorem. Algorithm 2 will choose many short sequences and a few long sequences (with exponential distribution of sequence length). Another option is a fixed size (like 5). Both have worked.

Rev6

> "this work may offer a useful alternative for continuous valued data"

This is a useful alternative for binary data as well, avoiding the difficulty in training RBMs (and even more, DBNs and DBMs) due to escaping the need to efficiently sample from the model in the inner loop of training (negative phase).

> "authors develop a new training algorithm for denoising autoencoders which they claim is better than existing methods - this is however not empirically validated"

The wallback algorithm is NOT at all a central contribution: it is an alternative way of training, justified by a theorem (we at least learn the same thing), the objective of getting rid of spurious modes faster, and experiments using both visual inspection of samples and log-likelihood bounds. We have added quantitative comparisons with and without walkback, as well as generated samples, showing its advantage.

> "the walkback algorithm in fact does not improve (quantitatively)"

We have modified the way to estimate the likelihood, making it a lower bound on the true likelihood. We have also better optimized the noise level with walkback. With these changes, we now find a significant and reliable difference in log-likelihood between the versions with and without walkback.

> "theoretical aspects of the paper are not sufficiently empirically validated"
> "ability to draw samples ... is very neat but it would be useful to have some significant empirical justification of the practical utility"

The theorems do not say that the walkback will be better, only that it learns the same thing. This has been validated. Our revised log-likelihood results clearly confirm our visual impression that walkback was better.
The main contribution is not the walkback algorithm, but the theorems and experiments showing that general denoising autoencoders estimate the data generating distribution and that samples can be obtained from them (WITH OR WITHOUT walkback).

> "Much remains to be solved to use DAEs as generative models"

Please clarify what you see as remaining to be solved. The core theorems apply to regular training of any parametrization of DAEs and corruption process. Probabilities can be estimated by non-parametrically running the Markov chain, simply by averaging the P(x_t=X |tilde{x_{t-1}}) collected along the chain for the value X of interest. This is how we obtain our likelihood bounds.

Rev7

> "While the experimental validation results are encouraging, they seem preliminary and not conclusive with respect to improvements over existing work"

This is the FIRST TIME one is successfully able to sample from a general denoising auto-encoder in a way that is theoretically justified. There is no claim made that this will necessarily be better than RBMs or other generative models, although the approaches are sufficiently different that it is worth exploring (in particular because some of the fundamental difficulties in training RBMs, the estimation of the negative-phase part of the gradient through sampling, are absent from the denoising auto-encoder framework).

> "in the abstract talks about sampling using MCMC … for what purpose?"

Being able to sample from a model is the cornerstone of probabilistic modeling. Using that, one can answer arbitrary questions about the variables modeled.

> "distinction between the C (i.e. C(X^tilde|X) and the P (i.e. P(X|X^tilde)?"

Just a notation device, because P is given by Nature and C is chosen by the person setting up the training. C stands for corruption.

> "How valid is the assumption that C(X^tilde|X) is known? This seems like a very restrictive assumption"

No, since C is CHOSEN by us. For example, C can be adding Gaussian noise or setting some variables to 0 like in dropout.

> "sampling from P(X|X_tilde) then from C(X_tilde|X)"

It should have been the other way around. Thanks for catching that.

> "How long to train"

About 50 epochs.

> "encouraging, but not conclusive with respect to prior work"

There is no prior work justifiably showing how to sample from general denoising auto-encoders.